# Generalized Telangiectasia as a Hallmark of Intravascular Lymphoma: A Case Report and Literature Review

**DOI:** 10.3390/healthcare12141455

**Published:** 2024-07-22

**Authors:** Weiwei Zhang, Ying Tang

**Affiliations:** 1Department of Endocrinology and Metabolism, The Second Affiliated Hospital of Chongqing Medical University, Chongqing 400010, China; zhangww@hospital.cqmu.edu.cn; 2Department of Pathology, West China Hospital, Sichuan University, Chengdu 610041, China

**Keywords:** intravascular lymphoma, telangiectasia, panniculitis, vasculitis

## Abstract

A 55-year-old woman had remarkably generalized telangiectasia. Large atypical lymphoid cells were found within the lumen of small vessels, and CD20 was positive in her third skin biopsy. She was diagnosed with intravascular large B-cell lymphoma (IVLBCL) and went into remission of the skin manifestations after seven courses of R-CHOP (rituximab, cyclophosphamide, doxorubicin, vincristine, and prednisolone) and four courses of intravenous high-dose methotrexate (HD MTX). To our knowledge, this is the first case report of IVLBCL with generalized telangiectasia in China. From a review of the literature, we found that generalized telangiectasia is a remarkable manifestation of intravascular lymphoma, and the differential diagnosis of intravascular lymphoma and panniculitis or vasculitis is very important. When the diagnosis is confusing, multiple skin biopsies are useful.

## 1. Introduction

Intravascular lymphoma (IVL) is a rare non-Hodgkin’s lymphoma. It was first described as endotheliomatosis and reticuloendotheliosis in 1959 because of the postulated endothelial origin [1]. A subsequent study found it was due to a malignant proliferation of lymphocytes [2]. About 85% of IVLs are B-cell lineages, which can also, though less common, originate from T cells or natural killer (NK) cells [3]. They are named intravascular large B-cell lymphoma (IVLBCL) and intravascular NK/T-cell lymphoma (IVNKTL), respectively. According to the latest WHO classification criteria for lymphoma, IVLBCL belongs to mature B-cell neoplasms, and IVNKTL is not recognized as a specific entity but is considered a form of extranodal NK/T-cell lymphoma (ENKTL) [4]. IVL is aggressive and disseminated for the proliferation of lymphoma cells in the lumen of small- to medium-sized blood vessels, particularly capillaries. Therefore, the presence of neoplastic lymphocytes in the lumen of small vessels is the condition sine qua non for the diagnosis of IVL [5]. Tumor cells can involve vessels of any organ in the body, resulting in highly variable and nonspecific symptoms and signs. Therefore, it is sometimes extremely difficult to make an accurate diagnosis. Some patients cannot receive the right diagnosis until an autopsy. Because of the late diagnosis, many patients die within a year after diagnosis. Chemotherapy is the main treatment method, but the prognosis is usually poor [6]. Early diagnosis and treatment are very important because, in this scenario, IVL may be curable with chemotherapy. Rituximab is an anti-CD20 chimeric monoclonal antibody. The addition of rituximab to conventional chemotherapies has a great advantage in the treatment of IVLBCL [5].

The clinical symptoms of IVL lack specificity. Its main clinical manifestations are skin lesions and symptoms of neurological involvement, often accompanied by fatigue, loss of weight, etc. The most common type of skin lesions are subcutaneous nodules and plaques accompanied by edema [7]. We report a case of a 55-year-old Chinese woman with IVLBCL who presented with remarkably generalized telangiectasia and was initially suspected to have panniculitis. Her diagnostic procedure reminds us that IVL can be confused with panniculitis, and generalized telangiectasia is a significant skin lesion of IVL, although it is rare. We also reviewed the literature regarding patients with typical generalized telangiectasia in IVL.

## 2. Case Presentation

A 55-year-old woman was admitted to the Department of Rheumatism and Immunology of our hospital with suspected panniculitis. She had experienced a fever for half a month and had edema of the thighs for one week. Her medical history was nothing special. Physical examination revealed obvious edema and diffuse cutaneous induration in both thighs and remarkable dendritic-like telangiectasia on the overlying skin (Figure 1A). Some subcutaneous indurative nodules could be palpated with no tenderness. She had no palpable hepatomegaly, splenomegaly, or lymphadenomegaly. A neurologic examination was unremarkable, and she had clear consciousness.

On admission, her body temperature showed a typical recurrent fever. The basal and highest body temperatures were 36.2 and 39.2 °C, respectively. The laboratory findings were hemoglobin of 98 g/L (reference range 115–150 g/L), lactic dehydrogenase of 3971 IU/L (reference range 110–220 IU/L), erythrocyte sedimentation rate (ESR) of 120 mm/h (reference range < 38 mm/h), C-reactive protein (CRP) of 137 mg/L (reference range < 5 mg/L), and serum β2-microglobulin of 2.74 mg/L (reference range 0.70–1.80 mg/L). Other laboratory tests were normal. She was found to have no evidence of infection. The bone marrow biopsy revealed active cell proliferation without abnormal cells. The enhanced magnetic resonance imaging (MRI) of the lower extremities revealed swelling and thickening of soft tissues in both thighs (Figure 1B). Chest computerized tomography (CT) showed a single low-density nodule in the left upper lobe of the lung. An abdominal CT scan found no splenomegaly or hepatomegaly. Brain CT and MRI revealed a mass (3.2 × 1.9 cm) on the left parietal lobe, which was possibly a meningioma, as its base was partly located close to the meninges.

One week after she was hospitalized, two biopsies of her indurated skin were conducted. The first one displayed mild skin keratosis. The second biopsy showed that the cutaneous small blood vessels were surrounded by a few small lymphocytes, which suggested vasculitis was possible. Then, she was given intravenous methylprednisolone 40 mg per day for anti-inflammatory treatment. After a week of treatment, her body temperature returned to normal, but the telangiectasia rapidly spread to the abdomen, chest, and neck, and several subcutaneous nodules appeared on the right arm. Therefore, a third skin biopsy was conducted. The specimen was purposefully taken from the site with superficial telangiectasia. This biopsy found some medium-sized lymphocytes within the vascular lumens of subcutaneous tissues (Figure 2A), and these cells were positive for CD20 (Figure 2B), CD138, and CD79a but negative for CD10, CD3, and CD56, suggesting IVLBCL. A lumbar puncture and another bone marrow biopsy were taken and yielded no evidence of tumor invasion, but the 18F-fluorodeoxyglucose (18F-FDG) positron emission tomography-computed tomography (PET-CT) showed increased uptake of 18F-FDG in the ethmoid sinus, nasal, and multiple bones, whereas the nodule in the left upper lobe of the lung showed unremarkable uptake.

Next, 4 weeks after admission, she was transferred to the hematology department for treatment. She was treated with the R-CHOP regimen (rituximab, cyclophosphamide, doxorubicin, vincristine, and prednisolone). Her telangiectasia regressed significantly after her first course of R-CHOP. But then she began to lose a lot of hair. Because of concerns about the side effects of the drugs, she failed to complete the second course of R-CHOP. Later, under the doctor’s persuasion, she accepted another five courses of R-CHOP completely. During treatment, an enhanced chest CT and another brain MRI were undertaken. The nodule in the left upper lobe of the lung remained unchanged. The brain MRI revealed that the mass was the same size as before. Taken together, the pulmonary nodule was likely due to chronic inflammation, and the mass on the brain was most probably a meningioma rather than a malignancy. She was still given intravenous high-dose methotrexate (HD MTX) for prophylactic chemotherapy. Finally, the patient went into remission from the skin manifestations after seven courses of R-CHOP and four courses of HD MTX. The follow-up will be continued.

## 3. Discussion

In recent years, an increasing number of cases have been reported to describe the characteristics of IVL, but the diagnosis and treatment of this malignancy are still difficult. Skin lesions are the common and maybe the initial manifestation of IVL. According to the subclassification of IVLBCL, if the disease is limited to the skin, the condition is called cutaneous variant and shows a better prognosis [8]. Due to our patient’s PET-CT showing increased uptake of 18F-FDG in multiple bones, this suggested that, in addition to the skin, there might also have been bone marrow infiltration. Therefore, cutaneous variants were not considered in this patient. Of all IVL cases reported, about 39% of patients have cutaneous symptoms [7,9]. They are described as nodules, plaques, and macules, often accompanied by edema. However, most of these skin lesions are atypical and also occur in other diseases, for example, panniculitis and vasculitis [10]. Therefore, patients with these skin lesions often initially seek help from a dermatologist or a rheumatologist. It is challenging to link these skin lesions with such an unusual disease, especially for doctors who are not hematologists.

This was not the first time an IVL patient with subcutaneous nodules was suspected to have panniculitis [11], as subcutaneous nodules are a frequent manifestation in both IVL and panniculitis. In some cases of IVL, these skin changes have been described directly as panniculitis-like lesions [12,13,14,15,16]. Other IVL cases have found that lymphoma cells can infiltrate subcutaneous adipose tissue with panniculitis formation [17,18]. Hence, panniculitis is an important differential diagnosis of IVL. Panniculitis is a disease of inflammation of the subcutaneous adipose tissue, and it can involve multiple organs to make its clinical manifestations diverse, just like IVL. Of note, the involvement of the CNS is scarce in panniculitis, which is common in IVL. Therefore, the neurological symptoms are particularly important in the differential diagnosis of these two disorders. The final diagnosis of these two diseases requires histological confirmation. Our patient suffered from a subcutaneous nodule but lacked neurological symptoms, so she was initially suspected of panniculitis. On the second skin biopsy, vasculitis was found. The vasculitis may be caused by the stimulation of neoplastic cells in cutaneous blood vessels. Quite a few IVL patients had vasculitis presentation in previous case reports [19,20,21,22,23,24,25,26], and several pathologic results confirmed that they had vasculitis [19,24,25]. A patient was misdiagnosed as having vasculitis [24]. Our patient was initially suspected of panniculitis. But the first and second skin biopsies of our patient revealed mild skin keratosis and vasculitis, respectively. No lymphoma cells were found, which may be related to the inappropriate site of these two skin biopsies. Therefore, the third skin biopsy was specifically taken from the site of telangiectasia, and the final histopathological examination revealed a large number of neoplastic lymphocytes with strong CD20 expression in the capillaries, confirming the diagnosis of IVLBCL. In general, when a patient presents with a panniculitis-like or vasculitis-like manifestation, IVL should be considered, and multiple skin biopsies are highly recommended, especially taking specimens from significant skin lesions.

Telangiectasia can appear in many diseases, but generalized telangiectasia is infrequent in IVL. To our knowledge, this is the first case of IVL with generalized telangiectasia. From a review of previous literature, we found another 24 IVL patients with similar skin lesions (Table 1), of which 22 cases were B-cell lineages, one case was T-cell lineage [27], and one case did not clarify the type of tumor cell [28]. Most patients were Caucasian; only six were from Asia [17,27,28,29,30,31]. The mechanism of generalized telangiectasia in IVL has not yet been fully illustrated, but the obstruction and recanalization of vessels in the subcutis are thought to be the main cause. Although IVL affects men and women equally, there was an obvious gender difference in IVL patients among the above-mentioned cases: 17 women and 7 men. It seems that women are more likely to have telangiectasia in IVL. The telangiectasia in the 24 cases almost always started from the legs and then gradually extended to the trunk, even to the face. On the whole, telangiectasia in IVL was generalized, not local. The telangiectasia does not occur solely. Induration or plaques of skin were associated lesions in nearly half of the patients [17,27,32,33,34,35,36,37,38], and edema was also an added symptom in 13 of the patients [14,15,17,24,32,33,35,39,40,41,42,43,44]. Noticeably, the skin lesions of three patients were described as mainly panniculitis-like [14,15,16], one patient had a lupus erythematosus skin lesion [45], and another two patients had orange-like skin changes [14,42]. Eight patients with neurological symptoms were reported, indicating tumor invasion of the nervous system. In addition to these symptoms, B symptoms, including fever and weight loss, were almost always present in the 24 patients, which are associated with lymphoma. In short, it is obvious that generalized telangiectasia is an unusual but remarkable manifestation of IVL, and it is often accompanied by skin induration, plaques, and edema. It is worth noting that among the 24 IVL patients we reviewed, many of their pictures looked overweight or even obese, and most of them were over 50 years old. Some studies have found that overweight/obesity and unhealthy lifestyles are risk factors for non-Hodgkin lymphoma [46,47]. This view seems to be confirmed in these patients.

The differential diagnosis of generalized telangiectasia mainly includes generalized essential telangiectasia (GET), ataxia telangiectasia (A-T), hereditary hemorrhagic telangiectasia (HHT), and cutaneous collagenous vasculopathy. GET is often asymptomatic and without other skin lesions except telangiectasia [48]. A-T is an autosomal recessive disorder that mostly has infant onset, with the telangiectasia initially appearing on the sclera [49]. HHT is an autosomal dominant disorder known as cutaneous and mucosal telangiectasia. HHT patients have a positive family history and recurrent episodes of bleeding with visceral involvement [50]. Cutaneous collagenous vasculopathy was first described in 2000 [51], and only a few patients have been reported by now. It mainly affects males without systematic symptoms except telangiectasia. Collagen deposition within the vascular walls of superficial cutaneous vessels is the pathogeny [52]. Generalized telangiectasia can also appear with a low probability in acquired immunodeficiency syndrome (AIDS) and mastocytosis. Telangiectasia was first observed in patients with AIDS in 1986 [53]. Later, three AIDS patients were also reported [54,55,56]. The telangiectasia in AIDS is linear and crescent-like, strikingly different from the telangiectasia in IVL, which is mainly dendritic-like. In mastocytosis, the telangiectasia is named telangiectasia macularis eruptive perstans (TMEP), characterized by a persistent, asymptomatic eruption of macules with telangiectasia [57]. TMEP accounts for less than 1% of mastocytosis, and an increased number of mast cells in the skin is reliable for the diagnosis. Clinically, the most frequent telangiectasia is spider angiomata, which is caused by elevated estrogen levels and mainly presents in patients with liver cirrhosis. It often occurs on the upper trunk and face with abnormal liver functions. In summary, it is not easy to make the right diagnosis for a patient with generalized telangiectasia, but when the diagnosis is confusing, multiple skin biopsies are useful.

## 4. Conclusions

In all, we report the first Chinese case of IVL with generalized telangiectasia as the highest manifestation. This case highlights the importance of a differential diagnosis of IVL with panniculitis or vasculitis. According to our patient and other similar reported cases, we believe that generalized telangiectasia is an unusual but remarkable manifestation of IVL. Specimens taken from the skin of telangiectasia are particularly useful for diagnosis. Multiple biopsies are recommended when the result is equivocal, especially for patients with atypical skin lesions. Multidisciplinary collaboration is also important for the diagnosis and treatment of IVL.

## Figures and Tables

**Figure 1 healthcare-12-01455-f001:**
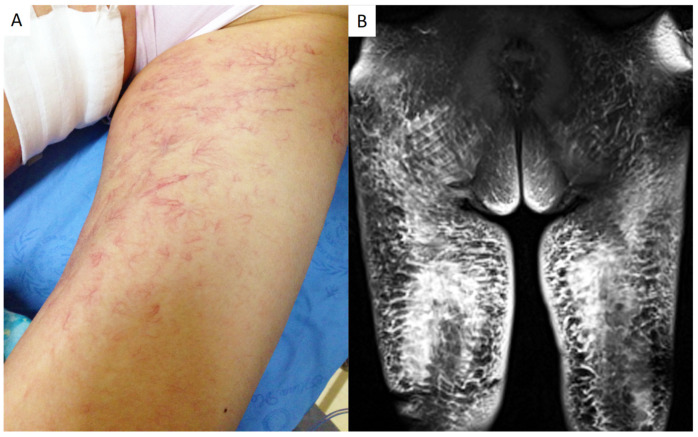
Telangiectasias overlying normal skin of the thigh (**A**); Obvious microvascular dilatation in both thighs on the image of T1-weighted contrast-enhanced magnetic resonance imaging (MRI) (**B**).

**Figure 2 healthcare-12-01455-f002:**
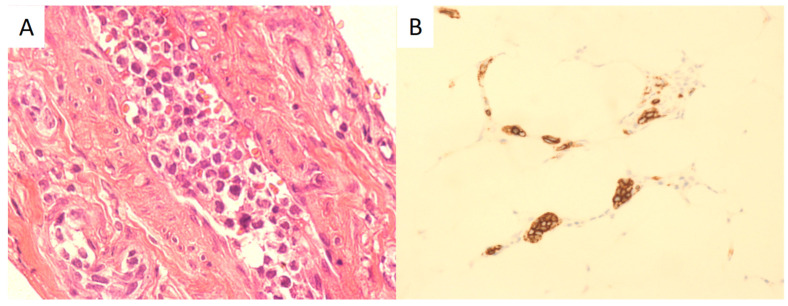
Skin biopsy specimen. Dilated vessels in the subcutaneous tissue occupied by large hyperchromatic cells (H&E, ×400) (**A**) and large hyperchromatic cells showing positive immunohistochemical staining with the CD20 B-cell marker (×200) (**B**).

**Table 1 healthcare-12-01455-t001:** Summary of 24 reported cases with intravascular lymphoma manifested by telangiectasia.

Cases	Nation	Age Sex	Type	Sites of Telangiectasia	Other Skin Lesions	Neurological Symptoms	Therapy	Outcome
Jang et al. [27]	Korea	23/F	T	breast, abdomen, back, extremities	erythematous patches	No	CHOP	Death
Cathebras et al. [39]	France	75/F	B	generalized	edema	No	N/A	N/A
Niiyama et al. [29]	Japan	72/F	B	entire body excepting face	N/A	No	R-CHOP	CR
Szuba et al. [40]	Poland	79/F	B	breasts, trunk, extremities,	edema	No	corticosteroids	Death
Vercambre-Darras et al. [32]	France	79/M	B	trunk, upper arms, thighs	induration, edema	No	R-CHOP	PR
Saleh et al. [41]	Lebanon	58/M	B	entire body	edema	No	N/A	Death
Barnett et al. [33]	America	68/F	B	lower extremities, breasts, abdomen	plaques, ulceration, edema	Yes	R-CHOP	N/A
Takacs et al. [42]	Hungary	79/F	B	chest, abdomen, lower extremities	orange-like skin, edema	Yes	R-CHOP	CR
Nishiyama et al. [17]	Japan	67/F	B	generalized	indurated plaques, edema	Yes	CHOP	PR
Wolter et al. [30]	Vietnam	64/F	B	trunk, lower limbs	N/A	Yes	corticosteroids	Death
Eros et al. [14]	Hungary	74/F	B	entire body	panniculitis-like lesions, orange-like skin, edema	Yes	PUVA therapy	CR
Eto et al. [31]	Japan	73/F	B	generalized	N/A	Yes	chemotherapy	Death
Sanna et al. [15]	Switzerland	84/F	B	widespread	panniculities-like lymphedema	N/A	polychemotherapy	CR
Ozguroglu et al. [34]	turkey	60/F	B	trunk	violaceous, infiltrated plaques	No	CHOP	PR
Walker et al. [24]	Germany	46/F	B	face, trunk	edema	Yes	cyclophospha	Death
Perniciarc et al. [43]	America	84/M	B	legs, thighs	redness of the legs, edema	No	CHOP	PR
Wilson et al. [35]	America	60/F	B	abdomen, thighs	induration of skin, edema	No	N/A	N/A
Weichert et al. [36]	Canada	73/M	B	lower abdomen, thighs	induration of skin	No	CHOP	C
Wahie et al. [37]	England	59/F	B	flanks, tights, lower abdomen	painful, indurated erythematous rash	No	R-CHOP	Death
Matsue et al. [28]	Japan	73/M	N/A	N/A	N/A	No	N/A	N/A
Han et al. [38]	America	81/M	B	bilateral thighs, waist, lower back	eruptions, patches, plaques	No	rituximab	CR
Sanchez-Cano et al. [45]	Spain	60/F	B	neck, thorax, back	lupus erythematosus skin lesion	Yes	R-CHOP	Death
Chuffart et al. [16]	France	80/F	B	thighs, abdomen	panniculitis-like skin lesions	N/A	R-CHOP	PR
Johanna et al. [44]	Spain	91/F	B	chest , back, abdomen	edema	N/A	N/A	PR

CHOP, cyclophosphamide, doxorubicin, vincristine, and prednisolone; R-CHOP, rituximab, cyclophosphamide, doxorubicin, vincristine, and prednisolone; PUVA, psoralen with ultraviolet-A; N/A, data not available, the study not performed; CR, complete remission; PR, partial remission.

## Data Availability

Data are contained within the article.

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
