# Peer review of "Generalized Telangiectasia as a Hallmark of Intravascular Lymphoma: A Case Report and Literature Review"

_healthcare, 2024, doi:10.3390/healthcare12141455_

Round 1

Reviewer 1 Report

Comments and Suggestions for Authors

The related manuscript is on a case report of generalized telangiectasia associated with VLBCL. No problems have been identified regarding the content and layout of the manuscript. In terms of case profile, the majority of patients are over 50 years of age. However, when the references given by the authors are examined, the pictures of the most cases found are overweight. Irregular and unhealthy eating patterns in older ages may cause the relevant disease. There are some studies on this situation. As advice, it would be better in terms of content if the authors conveyed their research and any observations regarding this matter.

Author Response

Comments: In terms of case profile, the majority of patients are over 50 years of age. However, when the references given by the authors are examined, the pictures of the most cases found are overweight. Irregular and unhealthy eating patterns in older ages may cause the relevant disease. There are some studies on this situation. As advice, it would be better in terms of content if the authors conveyed their research and any observations regarding this matter.

Response:Thanks for your comment. We admire the care and rigor with which you reviewed our manuscript. We have not previously focused specifically on the relationship between intravascular lymphoma and body weight. At your reminder, we reviewed the pictures of the 24 IVL patients with generalized telangiectasia and found that some patients did look overweight or obese, and most of the patients were over 50 years old. Our patient is also overweight. Indeed, a large number of studies have found that obesity and unhealthy lifestyle are risk factors for non-Hodgkin lymphoma. Therefore, we have added these concepts to the “Discussion” section of our manuscript and and added 2 relevant references. The content is as follows: “It is worth noting that among 24 IVL patients we reviewed, many of their pictures looked overweight or even obese, and most of them were over 50 years old. Some studies have found that overweight/obesity and unhealthy lifestyles are risk factors for non-Hodgkin lymphoma [51, 52]. This view seems to be confirmed in these patients”. Thank you again for your comments, which make our manuscript more scientific and valuable.

Reviewer 2 Report

Comments and Suggestions for Authors

I have carefully read and evaluated the case report. Here are my concerns and suggestions.

After mentioning intravenous B-cell lymphoma historically in the introduction, it would be more accurate to refer to the latest WHO and ICC classifications which were published in 2022.

Why did the patient get a seven-course of R-CHOP regimen the standard application is six cycles of R-CHOP.

Why is there no one in the hematology department in the case report? Was the patient treated in the endocrinology and metabolism unit?

There should be more references about the differential diagnosis of telangiectasias. For example there Is no reference for HHT. Some references must be deleted and some must be added.

Comments on the Quality of English Language

English is OK.

Author Response

Comments 1: After mentioning intravenous B-cell lymphoma historically in the introduction, it would be more accurate to refer to the latest WHO and ICC classifications which were published in 2022. 

Response 1:Thank you for pointing out this problem in manuscript. We have added the latest WHO classification of intravascular large B-cell lymphoma and intravascular NK/T-cell lymphoma in the “Introduction” section of our manuscript and cited relevant reference. The added content is as follows: “About 85% of IVLs are B-cell lineage, which could also, though less common, originate from T-cell or natural killer (NK)-cell [3]. They are named intravascular large B cell lymphoma (IVLBCL) and intravascular NK/T-cell lymphoma (IVNKTL), respectively. According to the latest WHO classification criteria for lymphoma, IVLBCL belongs to mature B-cell neoplasms,and IVNKTL is not recognized as a specific entity, but considered a form of extranodal NK/T-cell lymphoma (ENKTL) [4].” . This change will make the manuscript more rigorous. Thank you again!

Comments 2: Why did the patient get a seven-course of R-CHOP regimen the standard application is six cycles of R-CHOP. 

Response 2:We are very sorry for our negligence of this. Because the original focus of our manuscript was on the diagnosis, the treatment process was not described clearly. After our patient’s first course of R-CHOP, her telangiectasia regressed significantly. But she subsequently experienced a lot of hair loss, which made her very frustrated. Therefore, she dropped out of the treatment midway during the second course of R-CHOP. Finally, with the doctor's persuasion, she completed the subsequent 5 courses of R-CHOP. Although the patient received a total of 7 treatments, the complete treatment of R-CHOP was only 6 courses. In order to describe the treatment process more clearly and avoid misunderstanding, we added the following content in the “Case Presentation” section of our manuscript:“But then, she began to lose a lot of hair. Because of concerns about the side effects of the drugs, she failed to complete the second course of R-CHOP. Later, under the doctor's persuasion, she accepted another 5 courses of R-CHOP completely”. Thank you for your careful reminder, which helped us avoid a major omission.

Comments 3: Why is there no one in the hematology department in the case report? Was the patient treated in the endocrinology and metabolism unit?

Response 3:Thanks for your question. We totally understand your concern. It really need our explanation. The patient was first treated by the author Dr. Zhang Weiwei, when she was working in the department of Rheumatology and Immunology as an internal medicine resident at West China Hospital. The patient's typical manifestation is generalized telangiectasia, which is relatively difficult to identify the pathogeny. By consulting a large amount of literature, Dr. Zhang Weiwei considered that the pathogeny might be intravascular lymphoma, and recommended multiple skin biopsies for the diagnosis. Dr. Tang Ying from the department of pathology assisted in the pathological diagnosis of this patient. After being diagnosed with intravascular large B-cell lymphoma in the department of rheumatology and immunology, the patient was transferred to the department of hematology for treatment. We have stated this in the “Case Presentation” of our manuscript briefly. Because the main content of our manuscript was to discuss the patient’s skin changes and diagnosis, and her treatment process was only briefly mentioned. Therefore, there was no doctor from the department of hematology participated in the writing of this manuscript. Now, after completing the standardized training for internal medicine resident, Dr. Zhang Weiwei works in the department of endocrinology and metabolism at the Second Affiliated Hospital of Chongqing Medical University. As of now, Dr. Zhang Weiwei is still conducting telephone follow-up with the patient. I hope our answer can dispel your doubts. Thanks again for your comments.

Comments 4: There should be more references about the differential diagnosis of telangiectasias. For example there Is no reference for HHT. Some references must be deleted and some must be added.

Response 4:After your reminder, we have reorganized the references again. In the "Introduction" section of our manuscript, 2 references were deleted and 3 new references were added. In the “Discussion” section of our manuscript, we added 5 new references, including the references on ataxia telangiectasia (A-T) and hereditary hemorrhagic telangiectasia (HHT). Your suggestions helped us improve the scientific quality of the manuscript. Thank you very much!

Reviewer 3 Report

Comments and Suggestions for Authors

The manuscript describes a patient with Intravascular lymphoma (IVL) which is a rare and aggressive non-Hodgkin lymphoma type. A 55-year-old woman with intravascular large B-cell lymphoma presenting with generalized telangiectasia, initially misdiagnosed as panniculitis, emphasizing the diagnostic challenges of IVL.

To improve:

Clearly state the timeline of symptoms and their progression.

May be to emphasize on need of multiple biopsies to reach the final diagnosis.

Prognosis and long-term management of IVL patients to be mentioned/discussed.

Shorten discussion which includes repetitions.

The conclusion could be strengthened by including a brief overview of the implications for clinical practice, such as the importance of considering IVL in patients with atypical skin lesions and the need for multidisciplinary collaboration in diagnosis and treatment.

Author Response

Comments 1: Clearly state the timeline of symptoms and their progression.

Response 1:We gratefully appreciate for your valuable suggestion. The patient's symptom progressions were as follows. She began to have fever half a month before admission. Her lower limb edema appeared one week before admission. In the first week of hospitalization, she completed some blood tests and imaging examinations. In the second week of hospitalization, she underwent two skin biopsies and bone marrow biopsy. Because the second skin biopsy suggested the possibility of vasculitis, she was given a week of glucocorticoid treatment. But her telangiectasia did not improve with this treatment. So, in the third week of hospitalization, she underwent the third skin biopsy, which confirmed her diagnosis of intravascular large B-cell lymphoma. Next, in the fourth week of hospitalization, she was transferred to the hematology department for treatment. In our manuscript, we did not write a separate paragraph to describe this process, but wrote several paragraphs to describe her symptom changes and diagnostic progress. We wrote the time points in several content, but not completely. In order to show the timeline more clearly, we added two time points in the "Case Presentation" section of our manuscript. The first one is in the first sentence of the third paragraph, " Two biopsies from her indurated skin were conducted" was changed to “One week after she was hospitalized, two biopsies from her indurated skin were conducted”. The second one is in the first sentence of the fourth paragraph, " Next, she was transferred to the hematology department for treatment" was changed to " Next, 4 weeks after admission, she was transferred to the hematology department for treatment". Thank you for this valuable suggestion again! We hope that this revision will help readers better understand.

Comments 2: May be to emphasize on need of multiple biopsies to reach the final diagnosis.

Response 2:Skin biopsies are key to the diagnosis of our patient. It’s importance should be emphasized as you suggest. So, in the “Discussion” section of our manuscript, we added content is as follows: “Our patient was initially suspected of panniculitis. But the first and second skin biopsies of our patient revealed mild skin keratosis and vasculitis possible, respectively. No lymphoma cells were found, which may be related to the inappropriate site of these tow skin biopsies.. Therefore, the third skin biopsy was specifically taken from the site of telangiectasia, and the final histopathological examination revealed a large number of neoplastic lymphocytes with strong CD20 expression in the capillaries, confirming the diagnosis of IVLBCL”. We hope that these additional content can help readers understand why our patients need to undergo multiple skin biopsies.

Comments 3: Prognosis and long-term management of IVL patients to be mentioned/discussed.

Response 3:Thank you for pointing out this problem in manuscript. As this manuscript focuses primarily on the diagnostic process, we regret that we have written less about treatment. According with your advice, we have added and modified content in the first paragraph of “Introduction” section, as follows: “Because of the late diagnosis, many patients died within a year after diagnosis. Chemotherapy is the main treatment method, but the prognosis was usually poor [6]. Early diagnosis and treatment is very important, since in this scenario, IVL may be curable with chemotherapy. Rituximab is an anti-CD20 chimeric monoclonal antibody. Addition of rituximab to conventional chemotherapies has a great advantage in the treatment of IVLBCL [5]”. We hope these changes make our manuscript more reasonable. Thank you for your careful advice again!

Comment 4: Shorten discussion which includes repetitions.

Response 4:Thank you for your reminder. We did make an oversight. In the “Introduction” section of our manuscript, we repeated a sentence as follows: “IVL is aggressive and disseminated for proliferation of lymphoma cells in the lumen of small to medium-sized blood vessels”. We have deleted this duplicate sentence. In the “Discussion” section of our manuscript, this sentence “It has been more than 50 years since the IVL was first detected and defined” is also deleted. Because this sentence has a similar meaning to “It was first described as endotheliomatosis and reticuloendotheliosis in 1959 because of the postulated endothelial origin”,which is locates in "Introduction" section.

Comments 5: The conclusion could be strengthened by including a brief overview of the implications for clinical practice, such as the importance of considering IVL in patients with atypical skin lesions and the need for multidisciplinary collaboration in diagnosis and treatment.

Response 5:We revised the “Conclusion” section of our manuscript to strengthen the impact on clinical practice. The sentence “Specimens taken from skin of telangiectasia is particularly useful for diagnosis. Multiple biopsies are recommended when the result is equivocal” is modified as follows: “Specimens taken from skin of telangiectasia is particularly useful for diagnosis. In clinical practice, multiple biopsies are recommended when the result is equivocal, especially for patients with atypical skin lesions. The multidisciplinary collaboration is also important for the diagnosis and treatment of IVL”.

Reviewer 4 Report

Comments and Suggestions for Authors

The authors attempt to convey through case reports the importance of considering intravascular lymphoma as a differential diagnosis when encountering a patient with systemic telangiectasia.

Intravascular lymphoma is a subgroup of lymphoma, and although there are globally accepted review articles (J Clin Oncol. 2007; 25: 3168-73. , Lancet Oncol. 2009; 10: 895-902. )and reference books (WHO Blue book 4th edition) on the definition of the type, diagnosis, and clinical features, they are not cited and only older literature and case reports are cited. Please cite the literature again and show the reader what information needs to be confirmed in order to diagnose intravascular lymphoma.

In this case, the patient underwent a total of three skin biopsies before the diagnosis was made. It would be beneficial to the reader if you could explain in detail why the diagnosis could not be made after the first and second biopsies. Please provide the details and include them in the discussion.

The author notes that the majority of patients with systemic telangiectasia are female, suggesting a possible involvement of female hormones. While this is a unique consideration, I would like to see more scientific rationale for it. It may misinform the reader.

Author Response

Comments 1: Intravascular lymphoma is a subgroup of lymphoma, and although there are globally accepted review articles (J Clin Oncol. 2007; 25: 3168-73. , Lancet Oncol. 2009; 10: 895-902. )and reference books (WHO Blue book 4th edition) on the definition of the type, diagnosis, and clinical features, they are not cited and only older literature and case reports are cited. Please cite the literature again and show the reader what information needs to be confirmed in order to diagnose intravascular lymphoma.  

Response 1:Thank you for providing us with these excellent literature. Based on your suggestion, we added the citations of this reference and deleted two older references in the introduction of our manuscript. We also added content about the latest WHO classification criteria and the condition sine qua non for the diagnosis of intravascular lymphoma. To facilitate your review, we copy the revised content of this paragraph,as follows:

Intravascular lymphoma (IVL) is a rare non-Hodgkin's lymphoma. It was first de-scribed as endotheliomatosis and reticuloendotheliosis in 1959 because of the postu-lated endothelial origin[1]. Subsequent study found it was due to a malignant prolifer-ation of lymphocytes [2]. About 85% of IVLs are B-cell lineage, which could also, though less common, originate from T-cell or natural killer (NK)-cell [3]. They are named intravascular large B cell lymphoma (IVLBCL) and intravascular NK/T-cell lymphoma (IVNKTL), respectively. According to the latest WHO classifica-tion criteria for lymphoma, IVLBCL belongs to mature B-cell neoplasms,and IVNKTL is not recognized as a specific entity, but considered a form of extranodal NK/T-cell lymphoma (ENKTL) [4]. IVL is aggressive and disseminated for proliferation of lymphoma cells in the lumen of small to medium-sized blood vessels, particularly capillaries. Therefore, the presence of neoplastic lymphocytes in the lu-men of small vessels is the condition sine qua non for the diagnosis of IVL [5].The tumor cells can involve vessels of any organ in the body, resulting in highly variable and nonspecific symptoms and signs. So, it is sometimes extremely difficult to make an accurate diagnosis. Some patients couldn't get right diagnoses until autopsy. Because of the late diagnosis, many patients died within a year after diagnosis. Chemotherapy is the main treatment method, but the prognosis was usually poor [6]. Early diagnosis and treatment is very important, since in this scenario, IVL may be curable with chemotherapy. Rituximab is an anti-CD20 chimeric monoclonal antibody. Addition of rituximab to conventional chemotherapies has a great advantage in the treatment of IVLBCL [5].

  1. Pfleger L, Tappeiner J. On the recognition of systematized endotheliomatosis of the cutaneous blood vessels (reticuloendotheliosis?). Der Hautarzt; Zeitschrift fur Dermatologie, Venerologie, und verwandte Gebiete. 1959; 10: 359-63.
  2. Sheibani K, Battifora H, Winberg CD, Burke JS, Ben-Ezra J, Ellinger GM, Quigley NJ, Fernandez BB, Morrow D, Rappaport H. Further evidence that malignant angioendotheliomatosis is an angiotropic large-cell lymphoma. New England Journal of Medicine. 1986; 314: 943-8.
  3. Reyes‐Castro M, Vega‐Memije E. Intravascular large cell lymphoma. International journal of dermatology. 2007; 46: 619-21.
  4. Alaggio R, Amador C, Anagnostopoulos I, et al. The 5th edition of the World Health Organization classification of haematolymphoid tumours: lymphoid neoplasms. Leukemia. 2022; 36: 1720-1748.
  5. Ponzoni M, Ferreri AJ, Campo E, Facchetti F, Mazzucchelli L, Yoshino T, Murase T, Pileri SA, Doglioni C, Zucca E, Cavalli F, Nakamura S. Definition, diagnosis, and management of intravascular large B-cell lymphoma: proposals and perspectives from an international consensus meeting. J Clin Oncol. 2007; 25: 3168-73.
  6. Shimada K, Kinoshita T, Naoe T, Nakamura S. Presentation and management of intravascular large B-cell lymphoma. Lancet Oncol. 2009; 10: 895-902.

Comments 2: In this case, the patient underwent a total of three skin biopsies before the diagnosis was made. It would be beneficial to the reader if you could explain in detail why the diagnosis could not be made after the first and second biopsies. Please provide the details and include them in the discussion.

Response 2:Thank you for your suggestion. In order to help readers understand the reason why our patient underwent three skin biopsies, we have added content in the “Discussion” section of our manuscript, as follows: “Our patient was initially suspected of panniculitis. But the first and second skin biopsies of our patient revealed mild skin keratosis and vasculitis possible, respectively. No lymphoma cells were found, which may be related to the inappropriate locations of the skin biopsy. Therefore, the third skin biopsy was specifically taken from the site of telangiectasia, and the final histopathological examination revealed a large number of neoplastic lymphocytes with strong CD20 expression in the capillaries, confirming the diagnosis of IVLBCL”.  We hope that this revision will help readers better understand.

Comments 3: The author notes that the majority of patients with systemic telangiectasia are female, suggesting a possible involvement of female hormones. While this is a unique consideration, I would like to see more scientific rationale for it. It may misinform the reader.

Response 3:Thank you for your reminder. We were not rigorous enough in writing this. It’s just our hypothesis, and there is currently no relevant evidence. In order to make the manuscript more rigorous, we have deleted the relevant description of this hypothesis. Thank you for this valuable suggestion again! 

Reviewer 5 Report

Comments and Suggestions for Authors

Although you describe an unusual form of presentation of an unusual type of lymphoma, the quality of the article is low. There is a misconception that IVLBCL is of B-cell origin. The references need to be revised to include more recent and relevant descriptions of this type of lymphoma (e.g., Ponzoni et al, Blood 2018). The case description is not accurate as the diagnostic data is sparse. In addition, it should explain whether non-cutaneous infiltration was excluded, as this could be a cutaneous form with better prognosis. Finally, the structure of the article needs to be revised.

Comments on the Quality of English Language

The quality of the English is low, with repetitive phrases, inaccurate descriptions and frequent errors. 

Author Response

Comments 1: Although you describe an unusual form of presentation of an unusual type of lymphoma, the quality of the article is low.

Response 1:Thank you very much for your comment. The generalized telangiectasia was the main manifestation of our patient. Her diagnosis was difficult and challenging. In order to clarify the diagnosis, she underwent three skin biopsies. Her diagnostic process is very interesting. It suggests that generalized telangiectasia may be a manifestation of intravascular lymphoma and it is very important to conduct multiple skin biopsies for the diagnosis. That is why we completed this manuscript, and we believe it could be helpful for clinical practice. Now we have carefully considered all comments from the reviewers and revised our manuscript accordingly. Our manuscript has been revised a lot, and we hope to get your approval.

Comments 2: There is a misconception that IVLBCL is of B-cell origin.

Response 2:Thank you for your suggestion. In order to better elucidate the origin and classification of intravascular lymphoma, we have added the latest WHO classification of intravascular large B-cell lymphoma and intravascular NK/T-cell lymphoma in the “Introduction” of our manuscript, as follows: “About 85% of IVLs are B-cell lineage, which could also, though less common, originate from T-cell or natural killer (NK)-cell [3]. They are named intravascular large B cell lymphoma (IVLBCL) and intravascular NK/T-cell lymphoma (IVNKTL), respectively. According to the latest WHO classification criteria for lymphoma, IVLBCL belongs to mature B-cell neoplasms,and IVNKTL is not recognized as a specific entity, but considered a form of extranodal NK/T-cell lymphoma (ENKTL)[4]” . We hope that this additional content will be helpful. Thank you again!

Comments 3: The references need to be revised to include more recent and relevant descriptions of this type of lymphoma (e.g., Ponzoni et al, Blood 2018).

Response 3:Thank you for providing key information on the reference, which is very helpful to us. We have revised the manuscript as your suggestion. In the "Introduction" section of our manuscript, 2 older references were deleted and 3 new recent references were added. In the “Discussion” section of our manuscript, we added 5 new references, including the one you recommended.

Comments 4: The case description is not accurate as the diagnostic data is sparse.

Response 4:Thank you for your comment. The diagnosis of intravascular lymphoma is difficult, because the lymphoma cells can involve vessels of any organ in the body, resulting in highly variable and nonspecific symptoms and signs. The clinical symptoms of intravascular lymphoma lack of specificity. Its diagnosis relies on the finding of a large number of lymphoma cells in the lumen of small vessels by pathological examination. Our patient's main clinical manifestation was generalized telangiectasia. We couldn't clarify the diagnosis just based on her skin changes and blood test results. Her third skin biopsy revealed a large number of neoplastic lymphocytes with strong CD20 expression in the capillaries. This confirmed the diagnosis of intravascular large B cell lymphoma. During her diagnostic process, we did our best to clarify her diagnosis. But there may still be things we overlooked. We will try to do better in the future.

Comments 5:  In addition, it should explain whether non-cutaneous infiltration was excluded, as this could be a cutaneous form with better prognosis. Finally, the structure of the article needs to be revised.

Response 5:We gratefully appreciate for your valuable suggestion. Whether this patient is a cutaneous variant of intravascular large B-cell lymphoma is worthy of discussion. Because her PET-CT showed increased uptake of 18F-FDG in multiple bones, this suggested the possibility of bone marrow infiltration by lymphoma cells. So she was not considered cutaneous variant of intravascular large B cell lymphoma. We have added this content in the first paragraph of the “Discussion” section of our manuscript, as follows: “According to the subclassification of IVLBCL, if the disease limited to the skin, the condition is called cutaneous variant and shows a better prognosis[8]. Due to our patient's PET-CT showing increased uptake of 18F-FDG in multiple bones, this suggested that in addition to the skin, there might also be bone marrow infiltration. Therefore, cutaneous variant was not considered in this patient”. In addition, we have made a lot of structural changes to the manuscript based on the comments of other reviewers. We hope to get your approval.

Comments 6: The quality of the English is low, with repetitive phrases, inaccurate descriptions and frequent errors. 

Response 6:We appologize for the language problems in the original manuscript. We rechecked the manuscript and indeed found duplicate sentences and some errors. The revisions have been marked in the new manuscript. We have done our best to improve the writing of English, including checking the grammar through the professional Grammarly software. Thank you very much for your comments and suggestions.

Round 2

Reviewer 2 Report

Comments and Suggestions for Authors

After the changes the manuscript has evolved.The authors have replied all the questions and authors made the necessary adjustuments.  I would be happy to accept it for publication in its current form.

Comments on the Quality of English Language

After the changes the manuscript has evolved.The authors have replied all the questions and authors made the necessary adjustuments.  I would be happy to accept it for publication in its current form.

Reviewer 5 Report

Comments and Suggestions for Authors

Dear authors,

Greetings for your work. In the new version of the manuscript you have resolved the aforementioned problems, so I do not consider that it requires further modifications.